# Impact of Left Atrial Appendage Morphology and Function on Thrombosis Risk in Acute Ischemic Stroke: Insights from Transesophageal Echocardiography

**DOI:** 10.3390/medsci13020063

**Published:** 2025-05-22

**Authors:** Dung N. Q. Nguyen, Dung Thuong Ho, Thanh N. H. Tran

**Affiliations:** 1Faculty of Medicine, University of Health Sciences, Viet Nam National University, Ho Chi Minh City 7000, Vietnam; dunghothuong@gmail.com (D.T.H.); tnhthanh.y2019@uhsvnu.edu.vn (T.N.H.T.); 2Thong Nhat Hospital, Ho Chi Minh City 7000, Vietnam

**Keywords:** left atrial appendage thrombosis, ischemic stroke, transesophageal echocardiography

## Abstract

**Objective:** This study aims to investigate the correlation between the morphological and functional characteristics of the left atrial appendage (LAA) and the incidence of thromboembolic events by transesophageal echocardiography (TEE) in patients with acute ischemic stroke. **Methods**: This cross-sectional study included 171 patients with acute ischemic stroke, running from November 2022 to September 2024. Transesophageal echocardiography was performed to evaluate the presence of LAA thrombus. Multivariable logistic regression analysis was performed to identify risk factors for LAA thrombus. **Results:** Of the 171 patients, 19 (11.1%) were found to have LAA thrombus. Multivariable logistic regression identified two independent predictors of LAA thrombus formation: (1) left atrial spontaneous echo contrast (OR = 8, 95% CI: 3–19, *p* < 0.001) and (2) atrial fibrillation (OR = 8, 95% CI: 1.057–76.095, *p* = 0.044). **Conclusions:** Left atrial spontaneous echo contrast and atrial fibrillation are independent predictors of LAA thrombus in patients with acute ischemic stroke. The use of transesophageal echocardiography for early detection of LAA thrombus may help improve treatment strategies and prevent recurrent strokes.

## 1. Introduction

Stroke is the leading cause of morbidity and disability and remains one of the most common causes of death worldwide. Ischemic stroke accounts for approximately 87% of all stroke cases, with highly diverse etiologies [1]. The two primary causes of acute ischemic stroke are (1) thrombosis originating from atherosclerotic plaques in the carotid-vertebral artery system and (2) thrombosis originating from the left atrial appendage (LAA). Notably, patients with acute ischemic stroke caused by LAA thrombi have the highest rates of mortality, disability, and neurological impairment compared to other ischemic stroke subtypes.

The left atrial appendage (LAA) is a structure of the left atrium that functions as a blood reservoir and plays a role in releasing bioactive substances that regulate fluid balance and intracardiac pressure. The LAA’s anatomically complex structure, with multiple lobes and branches, makes it the primary site of thrombus formation in the left atrium. This risk is particularly high in conditions with impaired contraction and reduced blood flow velocity, such as atrial fibrillation and valvular heart disease [2]. However, LAA thrombi may also develop in some patients without concurrent atrial fibrillation or in those without atrial fibrillation detected at the time of stroke onset [3]. In such cases, the underlying cause of stroke may be overlooked, potentially leading to suboptimal treatment and prognosis. Therefore, the accurate identification of the etiology of ischemic stroke, especially the presence of LAA thrombus, is critical for appropriate management.

Transesophageal echocardiography (TEE) is a highly sensitive and specific imaging modality, comparable to computed tomography (CT) and superior to transthoracic echocardiography for detecting LAA thrombi [2,4]. Additionally, TEE allows for comprehensive assessment of LAA morphology and function, as well as the detection of associated cardiac abnormalities. This enables a better understanding of thrombus formation mechanisms and risk stratification, ultimately guiding treatment decisions and improving prognoses [5,6]. Current guidelines emphasize the role of TEE in evaluating patients suspected of having an LAA thrombus in the setting of ischemic stroke [6,7,8]. Previous studies have identified distinct LAA morphologies, such as “chicken wing” and “non-chicken wing,” which may influence thrombus formation [9,10]. However, the results are not consistent across studies.

Based on this background, we hypothesize that LAA morphology and function are independently associated with thromboembolic risk in patients with acute ischemic stroke. This study aims to provide new insights into the predictive value of LAA morphology and function for thrombus formation in acute ischemic stroke patients.

## 2. Materials and Methods

### 2.1. Patient Selection

We conducted a single-center, prospective, cross-sectional study on 171 patients with acute ischemic stroke from November 2022 to September 2024. All patients underwent brain magnetic resonance imaging (MRI) or computed tomography (CT) to confirm the diagnosis of acute ischemic stroke. Patients diagnosed with acute ischemic stroke who underwent transesophageal echocardiography (TEE) were included. Exclusion criteria included prior LAA closure, history of stroke within the last 3 months, and incomplete echocardiographic data. Carotid artery Doppler ultrasound was performed on all ischemic stroke patients. To exclude an atherosclerotic origin, MRI or CT angiography were conducted in patients with carotid artery stenosis detected on Doppler ultrasound. Per hospital protocol, all stroke patients were evaluated by experienced neurologists upon admission. Routine Holter monitoring (24–48 h) was performed to detect potential arrhythmias, including atrial fibrillation. Clinical data were collected from the hospital’s electronic database. The study was approved by the Ethics Committee in Biomedical Research of Thong Nhat Hospital (No. 78/BB-BVTN).

### 2.2. Transesophageal Echocardiography Procedure

After an overnight fast for 8 h, all patients underwent transesophageal echocardiography (TEE) within one week of hospital admission to detect LAA thrombus, with consent obtained from both the patient and their family. TEE was performed by two experienced cardiologists with the patient in the left lateral decubitus position. For all patients, local anesthesia was administered to numb the pharyngeal area. During TEE, the LAA was examined for thrombus using four different views: aortic valve short-axis (0–50°), mitral commissural view (50–80°), two-chamber view (80–110°), and long-axis view (>110°). LAA morphology was independently assessed by two experienced cardiologists using standardized criteria. The level of agreement between the two observers was measured using Cohen’s kappa statistic, yielding a κ value of 0.76, which reflects substantial inter-observer reliability. This level of concordance supports the consistency and reproducibility of the morphological classification used in this study.

### 2.3. Variable Definitions

Acute ischemic stroke was defined based on clinical and imaging criteria using brain CT or MRI. Patients were diagnosed with acute ischemic stroke if neurological deficit symptoms occurred within 7 days of onset. LAA thrombus was defined as an echogenic mass attached to the wall of the left atrium or LAA, observed in at least two planes, with echogenicity and mobility distinct from myocardial texture. Left atrial spontaneous echo contrast (LASEC), characterized by an echogenic swirling pattern of blood flow, was graded from 0 to 4 according to the Fatkin standard [11]. LAA morphology was classified based on TEE images into four types following the criteria of Wang and Kimura: cauliflower, cactus, chicken wing, and windsock [12,13].

### 2.4. Statistical Analysis

All statistical analyses were performed using IBM SPSS software, version 22.0. The Kolmogorov–Smirnov test was used to assess the normality of variable distributions. Continuous quantitative variables are presented as mean ± standard deviation or median, while qualitative variables are expressed as frequency (n) and percentage (%). Group differences were analyzed using independent t-tests for normally distributed continuous variables, while the Mann–Whitney U test was applied to non-normally distributed continuous variables. The Chi-squared test was used for qualitative variables. To identify predictors of LAA thrombus, we first conducted a univariate analysis. Variables with *p* < 0.2 were included in the multivariate analysis. We then performed multivariate logistic regression to determine independent predictors of LAA thrombus.

## 3. Results

Our study has demonstrated that LASEC is an independent risk factor for the presence of left atrial appendage thrombus in patients with acute ischemic stroke, in addition to the previously established predictor, atrial fibrillation.

### 3.1. Population Characteristics

In this study, the mean age of participants was 64.3 ± 12.5 years, with 108 male patients accounting for 63.1% of the total. LAA thrombus was detected on TEE in 19 patients (11.1%). The study population was divided into two groups: patients with LAA thrombus (LAA thrombus-positive) and those without LAA thrombus (LAA thrombus-negative). Baseline characteristics of all patients are summarized in Table 1. The results showed that atrial fibrillation was a significant risk factor for LAA thrombus formation (*p* < 0.001). Additionally, the left atrial diameter was significantly larger in the thrombus group than in the non-thrombus group (*p* = 0.034).

### 3.2. Morphological and Functional Characteristics of the Left Atrial Appendage on Transesophageal Echocardiography

The morphological and functional characteristics of the LAA on TEE are summarized in Table 2. Among 171 patients, 23.4% had chicken wing morphology, 20.5% had windsock, 43.3% had cactus, and 12.9% had cauliflower morphology. Thrombus formation was observed predominantly in the cauliflower group, but the difference is not statistically significant (*p* = 0.079). The results indicate that LAA length, LASEC, and LAA flow velocity were associated with LAA thrombus formation in patients with acute ischemic stroke.

### 3.3. Risk Factors Associated with the Presence of Left Atrial Appendage Thrombus

We selected risk factors associated with LAA thrombus from the previous analyses, including left atrial diameter, LAA length, LASEC, LAA flow velocity, and atrial fibrillation, for univariate and multivariate logistic regression analyses (Table 3). After multivariate logistic regression, only two factors were independently associated with an increased incidence of LAA thrombus in patients with acute ischemic stroke. Patients with LASEC had a significantly higher risk of LAA thrombus (OR = 8; 95% CI: 3–19; *p* < 0.001). Similarly, patients with atrial fibrillation had a higher rate of LAA thrombus than those with sinus rhythm (OR = 8; 95% CI: 1.057–76.095; *p* = 0.044). Additionally, LAA flow velocity was significantly lower in the thrombus group than in the non-thrombus group; however, it was not identified as an independent predictor of LAA thrombus in the logistic regression model.

## 4. Discussion

In comparison with national stroke registry data from Vietnam and large-scale international datasets, such as the Get with The Guidelines—Stroke (GWTG-Stroke) registry in the United States, the demographic characteristics of our study population (particularly in terms of mean age and the prevalence of common comorbidities such as hypertension and diabetes) are broadly similar. However, we observed a slightly higher rate of hypertension among our patients, which may reflect differences in regional health profiles or referral patterns. Additionally, given that our study was conducted at a single center in Southern Vietnam, where the population is relatively ethnically homogenous, the external validity of our findings may be limited when applied to more diverse populations outside of Southeast Asia.

### 4.1. The Prevalence of Left Atrial Appendage Thrombus on Transesophageal Echocardiography in Patients with Acute Ischemic Stroke

Globally, the prevalence of LAA thrombus detected on TEE in patients with acute ischemic stroke ranges from 7% to 25%. In our study, LAA thrombus was identified in 19 out of 171 patients, corresponding to an incidence rate of 11.1%. Among these cases, 63.2% occurred in patients with atrial fibrillation, while 36.8% were found in patients with sinus rhythm. Compared to previous studies, our study reported a higher incidence of LAA thrombus. Research by Tufan Cinar documented an incidence of 9.3% [14], while a study by James Anaissie and colleagues recorded a rate of 4.1% [15], both lower than our findings. Although these studies also focused on patients with acute ischemic stroke, differences in atrial fibrillation prevalence and patient demographics likely contributed to variations in LAA thrombus incidence. In Tufan Cinar’s study, patients with atrial fibrillation were excluded, whereas in James Anaissie’s study, the atrial fibrillation rate was only 6.8%, lower than in our study, leading to a lower incidence of LAA thrombus.

Our study also found that 7 out of 152 patients in sinus rhythm (Figure 1) had LAA thrombus, accounting for 4.6%, despite comprehensive evaluations using surface electrocardiography (ECG) and 24 to 48 h Holter ECG monitoring. The occurrence of thrombus in patients with sinus rhythm has gained increasing attention, as recent studies suggest that LAA thrombus formation is not solely attributed to atrial fibrillation but is also influenced by other factors, such as ischemic atrial cardiomyopathy, LAA dysfunction, and LAA deformation [14,16]. Collectively, these factors contribute to a higher incidence of LAA thrombus in patients with sinus rhythm.

Additionally, it is possible that some cases of LAA thrombus in patients with sinus rhythm on surface ECG and 24 to 48 h Holter ECG were due to paroxysmal atrial fibrillation that did not occur during the monitoring period. Therefore, for patients with echocardiographic findings suggestive of atrial fibrillation—such as left atrial enlargement, LASEC, and LAA thrombus—extended Holter ECG monitoring should be considered to detect paroxysmal atrial fibrillation.

### 4.2. Morphological and Functional Characteristics of the Left Atrial Appendage on Transesophageal Echocardiography

The morphology of the left atrial appendage (LAA) has long been of interest due to its association with an increased risk of thrombus formation. However, studies have reported conflicting results regarding the thrombotic risk associated with different LAA shapes [17,18].

In our study, the most common LAA shape was the cactus type, accounting for 43.3%, while the least common was the cauliflower type, at 12.9%. Among patients with LAA thrombus, the cauliflower shape had the highest prevalence, whereas the chicken wing and windsock shapes had the lowest prevalence, at 21.1%. Our findings are consistent with those of Biase Luigi et al. [9], who reported that patients with a chicken-wing-shaped LAA had a lower thrombus rate than those with other LAA morphologies.

However, a study by Korhonen M et al. on patients with suspected cardioembolic stroke found that the chicken wing shape accounted for 23.4% and was more commonly observed in stroke patients, whereas the cactus shape was less prevalent [10]. Variations in study populations and methods for assessing LAA morphology have contributed to discrepancies in conclusions regarding the relationship between LAA shape, thrombus formation, and stroke risk.

In our analysis of the association between LAA morphology and thrombus formation, we found no significant correlation between LAA shape and the presence of thrombus (*p* = 0.079). This finding aligns with several other studies suggesting that LAA morphology is more strongly associated with an increased risk of ischemic stroke or reduced blood flow velocity through the LAA rather than directly influencing thrombus formation [19].

### 4.3. Risk Factors Associated with the Presence of Left Atrial Appendage Thrombus

Our study found that patients with LASEC had an 8-fold higher risk of LAA thrombus formation. This finding is consistent with previous research. In a study by Fatkin et al., patients with LASEC had a 13-fold higher risk of thrombus formation compared to those without LASEC [9]. Similarly, Black et al. reported a 5-fold increased risk of thrombus formation in patients with LASEC compared to those without LASEC [20].

LASEC is strongly associated with LAA thrombus formation, as it reflects blood stasis within the appendage—a key precursor to thrombus development [21]. Therefore, patients with LASEC are at a higher risk of thrombus formation and ischemic stroke, and its presence may also indicate underlying atrial fibrillation. Consequently, when LASEC is detected during TEE in stroke patients, physicians should carefully assess multiple imaging planes to detect thrombi and consider extended Holter ECG monitoring to identify paroxysmal atrial fibrillation.

Atrial fibrillation has long been recognized as an independent risk factor for ischemic stroke, making atrial fibrillation screening a crucial component of stroke management. In patients with atrial fibrillation, assessing stroke risk using the CHA_2_DS_2_-VASc score is essential for guiding appropriate stroke prevention strategies [19].

In our study, atrial fibrillation was found to be significantly more prevalent in patients with LAA thrombus compared to those without. Multivariate regression analysis further confirmed a strong association between atrial fibrillation and LAA thrombus formation. This finding is consistent with previous studies, which have reported a higher prevalence of LAA thrombus in atrial fibrillation patients than in non-atrial fibrillation patients, reinforcing atrial fibrillation as an independent risk factor for thrombus formation [22,23]. Atrial fibrillation promotes thrombus formation through mechanisms such as blood stasis, endothelial dysfunction, and coagulation activation. The LAA, with its inherently slow blood flow velocity and complex multi-lobed structure, serves as the primary site for thrombus development in atrial fibrillation patients.

## 5. Limitations of the Study

This study has several limitations. Although we identified independent predictors of LAA thrombus, a formal predictive model with performance metrics (AUC, sensitivity, specificity) was not developed due to limited sample size. Future studies with larger cohorts and external validation are needed to assess the clinical applicability of these markers. In addition, LAA morphology was assessed using 2D echocardiography, which has inherent limitations in accuracy. Future studies utilizing 3D TEE, cardiac MRI, or CT imaging are needed for a more precise evaluation. Lastly, larger, multicenter, randomized, and prospective studies are required to validate our findings.

## 6. Conclusions

In our study, the prevalence of LAA thrombus was 11.1%, with 4.6% of cases occurring in patients with sinus rhythm. The presence of LASEC and atrial fibrillation were identified as predictors of LAA thrombus in these patients. Therefore, transesophageal echocardiography should be performed in patients with acute ischemic stroke when a cardioembolic source is suspected, regardless of whether atrial fibrillation is present. This approach may help prevent recurrent strokes. Additionally, in patients undergoing transesophageal echocardiography, if LASEC is detected, a thorough evaluation for thrombus should be conducted using multiple imaging planes. Furthermore, prolonged Holter ECG monitoring should be considered for enhancing the detection of atrial fibrillation.

## Figures and Tables

**Figure 1 medsci-13-00063-f001:**
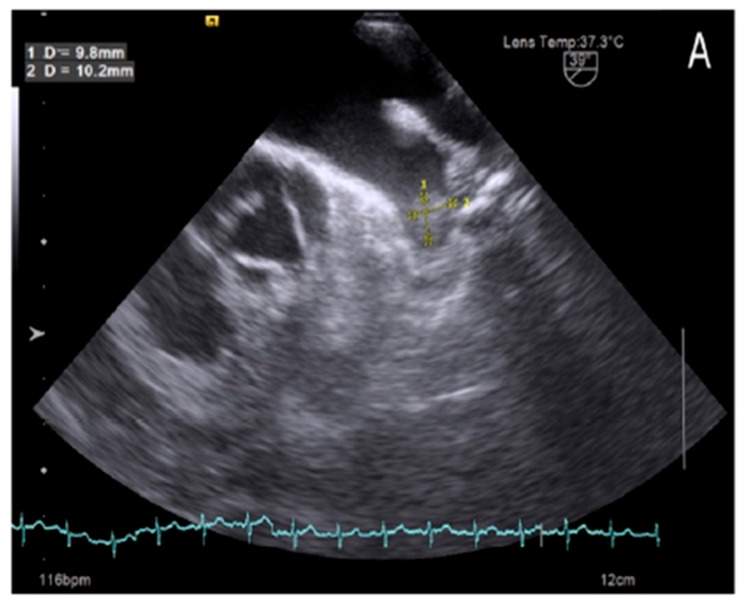
Left atrial appendage thrombus on transesophageal echocardiography images in patient with sinus rhythm.

**Table 1 medsci-13-00063-t001:** The baseline characteristics of all patients.

Characteristics	All Patients (n = 171)	Thrombus (+) (n = 19)	Thrombus (−) (n = 152)	*p* Value
Age, year	64.3 ± 12.5	65.0 ± 15.1	64.2 ± 12.2	0.705
Male, n (%)	108 (63.1)	12 (63.2)	96 (63.2)	1.000
AF, n (%)	18 (10.5)	12 (63.2)	6 (4.0)	<0.001
Hypertension, n (%)	124 (72.5)	13 (68.4)	111 (73.0)	0.673
Diabetes mellitus, n (%)	56 (32.7)	6 (31.6)	50 (36.5)	0.909
Dyslipidemia, n (%)	143 (83.6)	14 (73.7)	129 (84.9)	0.216
Coronary artery disease, n (%)	20 (11.7)	2 (10.5)	18 (11.8)	0.867
Heart failure, n (%)	4 (2.3)	0 (0.0)	4 (2.6)	0.476
LVEF, %	69.5 (65.3−75.7)	68.2 (59.5−77.4)	69.4 (59.6−79.2)	0.144
LVEDD, mm	46.0 ± 5.7	46.6 ± 5.3	46.0 ± 5.8	0.703
LA anterior–posterior diameter, mm	30.4 ± 4.6	34.3 ± 7.7	30.0 ± 3.9	0.034

Continuous variables are presented as mean ± SD or median; nominal variables are presented as frequency (%); AF, atrial fibrillation; LA, left atrium; LVEF, left ventricular ejection fraction; LVEDD, left ventricular end-diastolic diameter.

**Table 2 medsci-13-00063-t002:** Morphological and functional characteristics of the left atrial appendage on transesophageal echocardiography.

Characteristics	All Patients (n = 171)	Thrombus (+) (n = 19)	Thrombus (−) (n = 152)	*p* Value
Orifice size of LAA, mm	15.6 ± 3.7	17.1 ± 4.6	15.4 ± 3.6	0.135
Length of LAA, mm	30.0 ± 5.2	35.0 ± 5.0	29.4 ± 4.9	<0.001
Cactus, n (%)	74 (43.3)	5 (26.3)	69 (45.4)	0.079
Chicken wing, n (%)	40 (23.4)	4 (21.1)	36 (23.7)	
Cauliflower, n (%)	22 (12.9)	6 (31.6)	16 (10.5)	
Windsock, n (%)	35 (20.5)	4 (21.1)	31 (20.4)	
LASEC, n (%)	28 (16.4)	18 (94.7)	10 (6.6)	<0.001
Flow velocity of LAA, m/s	0.79 ± 0.28	0.55 ± 0.32	0.82 ± 0.26	0.002

LAA, left atrial appendage; LASEC, left atrial spontaneous echo contrast.

**Table 3 medsci-13-00063-t003:** Risk factors associated with the presence of left atrial appendage thrombus.

	Univariate Analysis		Multivariate Analysis	
	OR (95% CI)	*p* Value	OR (95% CI)	*p* Value
LA anterior–posterior diameter	1.00 (1.00–1.07)	0.082	-	-
Length of LAA, mm	1.00 (0.99–1.00)	0.064	-	-
Flow velocity of LAA	0.022 (0.003–0.036)	<0.001	2.0 (0.034−198.0)	0.068
LASEC	255.0 (30.0–2115.0)	<0.001	8.0 (3.0–19.0)	<0.001
AF	41.0 (12.082–144.024)	<0.001	8.0 (1.057–76.095)	0.044

LA, left atrial; LAA, left atrial appendage; LASEC, left atrial spontaneous echo contrast; AF, atrial fibrillation.

## Data Availability

The original data of this study will be made available upon reasonable request to the relevant researchers (Available online: https://docs.google.com/spreadsheets/d/1MrzxKCDZxUnknXerNY7NCFxYSZINpvpi/edit?gid=1625321709#gid=1625321709 (accessed on 20 February 2025)).

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
