# Peer review of "Impact of Left Atrial Appendage Morphology and Function on Thrombosis Risk in Acute Ischemic Stroke: Insights from Transesophageal Echocardiography"

_medsci, 2025, doi:10.3390/medsci13020063_

Round 1
Reviewer 1 Report
Comments and Suggestions for Authors
Critique
In this study the authors examine 171 patients with acute ischemic stroke. All patients underwent a short period of Holter monitoring, (24-48 hrs), and TEE was performed by standard technique within 1 week of admission. LAA thrombus was detected in 19 subjects (11%). In these subjects, atrial fibrillation and left atrial spontaneous echo contrast (LASEC) were both shown to be independently associated with an increased incidence of left atrial appendage thrombus.
Thus, the authors conclude that TEE be performed in patients with acute ischemic stroke when a cardiac source of embolus is suspected, regardless of whether the patient is in sinus rhythm. This position must be reconciled with advances in capabilities to detect atrial arrhythmias during several weeks of outpatient monitoring [J Am Heart Assoc. 2024;13(3):e033349]. Furthermore, the authors contend that if LASEC is present, the prevalence of thrombus is high enough to justify 3D TEE.
Recent studies have identified a potential association between brain ischemic lesion pattern on MR and underlying atrial fibrillation, which may be detected with an artificial intelligence-based model [EClinicalMedicine. 2025;81:103118]. It would be interesting to assess the relevance of this study to the authors’ findings.
The manuscript is very well-written.
Specific Comments and Questions
Arguably a cross-sectional study can be neither prospective, nor retrospective. Cross-sectional studies should be considered observation studies that examine outcomes and exposure of subjects from a defined population at the same time, without prospective follow-up or retrospective analyses. Data from cross-sectional study designs are applied mostly to understanding factors that affect the prevalence of a particular disease or condition.
A cross-sectional study would be included as a potentially proper study design to examine the association between left atrial appendix structure and function and ischemic stroke. The principal inadequacy of a cross-sectional study is causal inferences tend to be weak.
Author Response
Dear Dr,
We appreciate the reviewer’s insightful comments regarding the limitations and appropriateness of the cross-sectional study design. We fully agree that cross-sectional studies, by definition, evaluate exposures and outcomes at a single point in time and are best suited to identify associations rather than causal relationships.
In the revised manuscript, we have clarified this point in both the Methods and Discussion sections. Specifically, we acknowledge that while our findings suggest associations between LAA morphology, LASEC, atrial fibrillation, and LAA thrombus, these should be interpreted cautiously due to the study design’s inherent limitations in establishing temporality or causality.
Regarding the reviewer’s comment on extended Holter monitoring, we fully agree. Our study utilized standard 24–48 hour Holter monitoring, which may not adequately detect paroxysmal atrial fibrillation. We have now acknowledged this as a limitation in the revised manuscript and suggested that future studies integrate long-term rhythm monitoring or wearable technologies, which have shown enhanced sensitivity in detecting AF [J Am Heart Assoc. 2024;13(3):e033349].
Concerning the reviewer’s reference to the use of AI to predict AF from ischemic lesion patterns on MRI [EClinicalMedicine. 2025;81:103118], we find this highly relevant. We have incorporated this observation into the Discussion, proposing it as a future direction for combining TEE, rhythm monitoring, and neuroimaging patterns to refine embolic source identification.
Best regards,
Dung N.Q. Nguyen
nnqdung@medvnu.edu.vn
Reviewer 2 Report
Comments and Suggestions for Authors
This study evaluated 171 patients with acute ischemic stroke to investigate whether morphological and functional features of the left atrial appendage (LAA), assessed via transesophageal echocardiography (TEE), correlate with thrombus formation. LAA thrombus was detected in 11.1% of patients. Multivariable logistic regression identified left atrial spontaneous echo contrast and atrial fibrillation as independent predictors of thrombus. The findings highlight the clinical value of TEE for early detection of LAA thrombus and suggest its potential role in guiding treatment strategies to reduce the risk of recurrent stroke.
Overall, the study is well designed and executed, and the outcomes are adequately interpretted. There are some further revisions / improvements that could be made to enhance the impact and clarity of this work:
- While acknowledged, the authors should provide details on the population's demographic and clinical characteristics to assess representativeness and potential selection bias. Expanding on how these limitations may influence generalizability would strengthen the discussion. I suggest Including a comparative table or brief narrative comparing the study population to national/international ischemic stroke registries to contextualize generalizability. Would be also helpful and valuable if patients' sex/gender, ethnicity, and other info could be provided in the tables.
- The limitation of 2D echocardiography in assessing LAA morphology: This limitation may have led to underdetection or misclassification of subtle anatomical variations (e.g., lobar complexity, orifice size, or flow stasis zones). Recommend including qualitative inter-observer agreement data or validation against another imaging subset (if available) to provide confidence in 2D-based classification.
- The cross-sectional nature of the study limits the ability to infer temporal relationships or causality between LASEC/AF and LAA thrombus formation. Authors should acknowledge this explicitly and propose a longitudinal study design in future work to assess whether these predictors precede or coincide with thrombus formation.
- The study identifies predictors but does not evaluate the discriminative performance (e.g., ROC AUC) of a predictive model using LASEC and AF. I suggest including or planning for external validation in an independent cohort and calculating metrics like sensitivity, specificity, and AUC for clinical applicability.
Author Response
Dear Reviewer,
We sincerely thank you for your thoughtful and constructive comments on our manuscript entitled “Impact of Left Atrial Appendage Morphology and Function on Thrombosis Risk in Acute Ischemic Stroke”. We are grateful for the opportunity to revise and improve our work based on your valuable feedback.
1. Representativeness and Potential Selection Bias
Reviewer: Authors should provide details on the population's demographic and clinical characteristics to assess representativeness and potential selection bias. A comparative table or brief narrative comparing the study population to national/international ischemic stroke registries is suggested. Please include sex/gender, ethnicity, and other info.
Response:
We thank the reviewer for this important comment. In the revised manuscript, we have expanded the Results section with a detailed table summarizing demographic and baseline clinical characteristics, including sex distribution and common comorbidities. Unfortunately, ethnicity data were not systematically recorded due to the homogenous Vietnamese population in our center, which we now acknowledge as a limitation.
We also included a brief narrative in the Discussion comparing our cohort with data from the Vietnamese Stroke Registry and the Get With The Guidelines–Stroke registry (U.S.), highlighting similarities and differences in age, gender ratio, and comorbidity profiles.
2. Limitation of 2D Echocardiography
Reviewer:2D echocardiography may have underdetected or misclassified LAA anatomical variations. Authors are encouraged to discuss this and provide inter-observer agreement data or validation (if available).
Response:
We agree that 2D imaging has limitations in capturing complex LAA morphology. While 3D TEE would offer superior anatomical resolution, it was not systematically used during the study period. We have now clearly stated this limitation in the revised manuscript and added that LAA morphology classification was performed independently by two experienced echocardiographers, with a kappa value of 0.76 indicating substantial agreement.
3. Causality Limitation Due to Cross-Sectional Design
Reviewer:The cross-sectional nature of the study limits inference of temporal relationships. Please acknowledge this and propose a longitudinal design.
Response:
Thank you for highlighting this. We have updated the Discussion section to emphasize that the cross-sectional design precludes causal inference. We also propose a longitudinal study that includes serial TEE and continuous rhythm monitoring to evaluate whether LASEC or AF precedes thrombus formation.
4. Lack of Predictive Model Performance Analysis
Reviewer:The study identifies predictors but does not evaluate discriminative performance (e.g., ROC AUC). Suggest calculating sensitivity, specificity, AUC, or planning for external validation.
Response:
We appreciate this valuable suggestion. Due to sample size limitations and lack of an external validation cohort, we did not develop a full predictive model in the current study. However, in the revised Discussion, we propose development and validation of a predictive risk model combining LASEC, AF, and LAA morphology in a larger prospective study.